# Discordance for Potter’s Syndrome in a Dichorionic Diamniotic Twin Pregnancy—An Unusual Case Report

**DOI:** 10.3390/medicina56030109

**Published:** 2020-03-04

**Authors:** Stoyan Kostov, Stanislav Slavchev, Deyan Dzhenkov, Strahil Strashilov, Angel Yordanov

**Affiliations:** 1Department of Gynecology, Medical University Varna, 9000 Varna, Bulgaria; drstoqn.kostov@gmail.com (S.K.); st_slavchev@abv.bg (S.S.); 2Department of General and Clinical pathology, Forensic Medicine and Deontology, Medical University Varna, 9002 Varna, Bulgaria; dzhenkov@mail.bg; 3Department of Plastic Restorative, Reconstructive and Aesthetic Surgery, Medical University Pleven, 5800 Pleven, Bulgaria; dr.strashilov@gmail.com; 4Department of Gynecologic Oncology, Medical University Pleven, 5800 Pleven, Bulgaria

**Keywords:** potter’s sequence, dichorionic, oligohydramnios, extrarenal features, pulmonary hypoplasia

## Abstract

Introduction: Potter’s syndrome, also known as Potter’s sequence, is an uncommon and fatal disorder. Potter’s sequence in a multiple pregnancy is uncommon, and its frequency remains unknown. Worldwide in a diamniotic twin pregnancy, there are only a few cases described. Case report: We present an unusual case discordance for Potter’s syndrome in a dichorionic diamniotic twin pregnancy. Twin A had the typical physical and histological Potter’s findings. Twin B had normal respiratory function and normal physical examination findings. There are many controversies about this condition in diamniotic twin pregnancy. One case report concluded that that the presence of a normal co-twin in diamniotic pregnancy prevented the cutaneous features seen in Potter’s syndrome and ameliorated the pulmonary complications, whereas two other case studies reported that the affected twin had extrarenal features typical of the syndrome. Conclusion: We performed an autopsy and calculated lung weight/body weight ratio to diagnose pulmonary hypoplasia. Histopathologic examination of lungs and kidneys was performed. We concluded that the appearance of extrarenal features in the affected twin depends on the amniocity.

## 1. Introduction

Potter’s syndrome (PS) is an uncommon fatal disorder with an incidence of 1 in 4000 singleton pregnancies. Edith Potter first described it in 1946. The sequence is associated with bilateral renal agenesis, oligohydramnios, and pulmonary hypoplasia (PH). Renal abnormalities, which can include bilateral renal agenesis, severe hypoplasia, dysplasia, polycystic kidney, or obstructive uropathy, are the primary defect [1,2,3]. The incidence of PS in multiple pregnancies remains unknown. We report a case of PS in one of a twin pair in a dichorionic diamniotic twin pregnancy. Case studies suggest that in monoamniotic pregnancy, the affected twin has no extrarenal features of this syndrome, whereas, in diamniotic pregnancy, there are controversial reports. There are very few previous cases describing this condition in dichorionic twin pregnancy.

## 2. Case Report

A 21-year-old female multigravida (gravida 2, parity 2, abortion 0) at 34 weeks of twin pregnancy was admitted to our hospital with preterm prelabor rupture of membranes followed by uterine contractions. There was no history of smoking, alcohol, toxemia, exposure to drugs, radiation, or any diseases during pregnancy. The pregnancy was not booked and without regular follow up. There was no history of birth defects in her family members. The patient had had two previous singleton vaginal deliveries four and two years ago. The first vaginal delivery was at 35 weeks of pregnancy. The fetus was at cephalic presentation, female with birth weight—2055 g. The neonate had PS (absence of kidneys, lung hypoplasia, PS facies, congenital bilateral talipes equinovarus) and died twenty minutes after birth due to respiratory distress syndrome (Figure 1). 

A genetic study was refused. A lung weight/body weight ratio was not performed.

The second pregnancy went without any complications, and the baby was in good health condition. After hospitalization, ultrasound examination revealed diamniotic twin pregnancy with an absence of amniotic fluid in twin A and oligohydramnios in twin B. Twin A and twin B were in cephalic and in breech presentation, respectively. Both placentas were in fundal position and Grade II. Vaginal examination revealed cervical dilatation of 5 cm. Although the amniotic membrane in twin A was intact, leakage of amniotic fluid during examination was noticed. 

Twin A exhibited signs of fetal distress. External cardiotocography (CTG) showed variable decelerations. We decided to perform a cesarean section due to variable decelerations, anhydramnios, and prematurity of twins. Twin A had an intact amniotic membrane and an absence of amniotic fluid. The neonate had extrarenal manifestations of PS and growth weight—1690 g, length 32 cm, head circumference 31 cm. Twin A died 3 h after birth due to respiratory distress secondary to PH. The Apgar score at one and five minutes was two and three, respectively. There was a preterm prelabor rupture of membranes of twin B. This twin had a growth weight—1350 g, length—30 cm, head circumference—30 cm, and no visual anomalies. The Apgar score at one and five minutes was three and five, respectively. Twin B had a normal neonatal course and was discharged 14 days after delivery. Both babies were male with two separate placentas, amniotic sacs, and umbilical cords. Their umbilical cords contained two arteries and a vein. The pregnancy was dizygotic as neonates had different blood types—Twin A—“AB”/+/; Twin B—“A”/+/. There was an extremely small chance of mosaicism or blood chimerism, which is more likely to occur in monochorionic twins. An autopsy of twin A was performed (Figure 2). He had typical facial features for PS—a flattened nose, low set ears, a specific suborbital crease, and a short neck. Another facial anomaly was a preauricular sinus, which is not typical for this condition. Limb malformations were also present—congenital bilateral talipes equinovarus. Heart, liver, spleen, anus, and genitals were normal. Further examination revealed polycystic kidneys and lung hypoplasia—lung weight 14 grams. The calculated lung weight/body weight ratio (LW/BW) was 0.008. A histopathologic examination of lungs and kidneys was performed (Figure 3). 

## 3. Discussion

PS is also known as Potter’s sequence, because, despite differences in the urological abnormalities in individual cases, the endpoint is always a severe decrease or absence of amniotic fluid [2]. PS is an unusual fatal disorder that predominantly affects male babies and is common in primigravid mothers [2,3]. PS has been divided into four distinct subgroups (Table 1) [4,5].

As shown in the table, renal failure is the primary defect [3]. Renal anomalies lead to oligohydramnios. A decrease in the volume of amniotic fluid causes continuous pressure of the uterine wall, which leads to lung hypoplasia, face and limb abnormalities [3,4]. The degree of hypoplasia depends on the stage of lung development at which oligohydramnios occurs [3].

The manifestations of extrarenal features of PS depend on the amniocity. Case studies show that in monoamniotic twin pregnancy discordance for PS, the affected twin has no extrarenal features. In our opinion, the very few cases where we have a discordance for PS in diamniotic pregnancy are of particular interest as there is no consensus whether the unaffected twin protects the other by producing amniotic fluid and by decreasing the intrauterine pressure. 

McNamara reported two cases of monoamniotic pregnancies and PS in one of a twin pair. He discussed that one twin in each set had anomalies that, in a singleton or diamniotic pregnancy, would likely have resulted in fetal PH. He concluded that adequate amniotic fluid provided by the normal twin might prevent PH in the affected twin [6]. Mauer et al. described a case with PS in monoamniotic twin pregnancy. Twin A had an absence of kidneys but had none of the typical physical findings of PS other than low-set ears. He concluded that extrarenal manifestations of PS are due to oligohydramnios [7]. Cilento reported a case study with a monochorionic monoamniotic twin pregnancy. The male twin was born with no functional renal tissue and without the extrarenal manifestations of Potter facies, skin changes, club feet, and pulmonary hypoplasia. [8] Peres-Brayfield et al. observed an unusual case of a monoamniotic twin discordant for bilateral renal agenesis with normal pulmonary function [9]. We summarized monoamniotic case information in Table 2.

Based on their observations and conclusions, it is clear that in monoamniotic pregnancy with this condition, the amniotic fluid sharing can prevent respiratory distress syndrome due to providing the necessary environment for normal lung development [6,7,8,9,10]. Unfortunately, the mortality rates remain high due to renal or other anomalies associated with PS.

Fadel et al. observed a case with bilateral renal agenesis in one of a twin pair. The pregnancy was diamniotic, but the chorionicity was not described in the report. Antenatal ultrasound examination revealed that Twin A appeared to be normal, and Twin B had decreased amniotic fluid volume and lack of kidneys and bladder. Twin B died shortly after birth due to respiratory distress secondary to PH. Postnatal ultrasound examination showed the absence of kidneys. An autopsy was refused. They did not mention any extrarenal features of PS [11]. Sutherland reported a case with dichorionic twin gestation discordant for renal dysplasia and oligohydramnios. On ultrasound screening, the affected twin had bilateral hydroureteronephrosis, renal hyperechogenicity with cystic changes, oligohydramnios, and a massively dilated bladder. At delivery, he had no extrarenal features for PS—only minor PH and cutaneous abnormalities—very mild nasal and ear compression [11]. Holden et al. observed three cases of diamniotic twins, discordant for complete urinary tract obstruction and anhydramnios, with regular ultrasound scans and after delivery. Two affected twins died shortly after birth with confirmed severe PH. Selective feticide was performed in one of the cases [12]. We summarized diamniotic case information in Table 3.

As shown in the table, zygocity, LW/BW ratio, histopathological findings are reported only in the presented case. The limitations of this case study are that there were no prenatal ultrasonography and genetic examination at her first and third pregnancies. However, other cases also provide important findings. In his report, Sutherland concluded that the presence of a normal co-twin in diamniotic pregnancy prevented the cutaneous features seen in PS and ameliorated the pulmonary complications [11]. The other case studies reported that the neonate died shortly after birth due to respiratory distress syndrome [10,12]. Holden came to the conclusion that normal amniotic fluid volume in one amniotic sac does not protect the anhydramniotic twin from PH [12]. In our case, the affected twin had many extrarenal features of this syndrome—flattened nose, low set ears, specific suborbital crease, short neck, congenital bilateral talipes equinovarus, and hypoplastic lungs. The neonate was in constant intrauterine pressure due to oligohydramnios. Our case showed that in diamniotic twin pregnancy discordance for PS, the affected twin had severe PH.

In some cases, the cause of PS is genetic abnormalities: autosomal dominant or recessive inheritance of polycystic kidney disease, hereditary kidney dysplasia (caused by RET and UPK3A gene mutations) or chromosomal abnormalities. The syndrome occurs sporadically but may be inherited when the autosomal dominant triad causes it. PS often occurs in infants who have a family history of kidney abnormalities [3,13]. In our case, the patient had no relatives with kidney abnormalities, but her first baby was with PS. She refused genetic examination and consultation after her first and third delivery. Association between PS and other anomalies have been reported in the literature—Prune Belly anomaly, agenesis of corpus callosum, or VACTERL association. VACTERL association is an acronym for vertebral defects (V), anal atresia (A), cardiac malformations (C), tracheoesophageal fistula with esophageal atresia (TE), renal, radial dysplasia (R), and limb malformations (L) [14,15,16]. Antenatal diagnosis of PS is suspected when an ultrasound shows oligohydramnios and/or absence of bladder and kidney abnormalities [13,17]. Postnatal diagnosis of PH is suggested by calculating LW/BW ratio or low alveolar counts determined (RAC) by morphometric methods. LW/BW ratio of less than or equal to 0.012 or RAC of less than or equal to 4.1 (75% of mean normal count) are diagnostic criteria [18]. Paepe et al. estimated normal LW/BW for term and preterm infants. LW/BW at 28–36 gestational weeks is 0.025. LW/BW ratio at 37–41 gestational week is 0.017 [19]. There is no utilized consensus about treatment in normal and multiple pregnancies with PS. Neonates with PS have a deadly prognosis due to severe PH [12]. Serial amnioinfusion may improve outcomes but should be considered as innovation or research [20,21,22]. A vesicoamniotic shunt might be a choice of treatment in diamniotic pregnancy discordant for lower urinary tract obstruction. Unfortunately, opinions differ about the therapeutic value of shunting in twin pregnancy and it should be discussed with parents [23]. There is also controversy about whether selective feticide is an option in dichorionic pregnancy discordant for PS. A retrospective cohort study in 10 perinatal centers in the Netherlands over the period 2000–2010 was performed. The study involved women with dichorionic twin pregnancies discordant for structural or genetic abnormalities. Patients with twin pregnancy underwent fetal reduction to a singleton pregnancy [24]. Authors came to the same conclusion as Linskens et al., who recommended selective feticide only in a small number of cases of severe but non-lethal anomalies in dichorionic twins with PS [24,25]. As PS is a lethal anomaly, we can conclude that selective feticide in dichorionic pregnancy discordant for PS should be avoided. Pregnancy care and delivery do not change if the unaffected twin demonstrates normal test results [10,13].

## 4. Conclusions

PS in one of a twin pair in diamniotic twin pregnancy is an unusual condition. We come to the conclusion that the normal co-twin in diamniotic pregnancy does not prevent the extrarenal features seen in PS, and that this is possible only in a monoamniotc pregnancy [11].

## Figures and Tables

**Figure 1 medicina-56-00109-f001:**
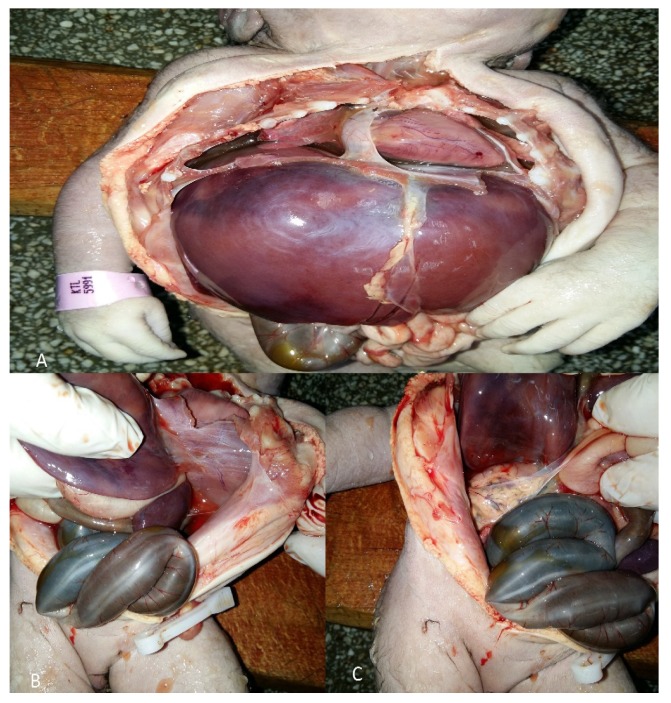
An autopsy of the first neonate with Potter’s syndrome (PS). The autopsy was performed 4 years ago: (**A**) Lung hypoplasia; (**B**) absence of left kidney; (**C**) absence of right kidney.

**Figure 2 medicina-56-00109-f002:**
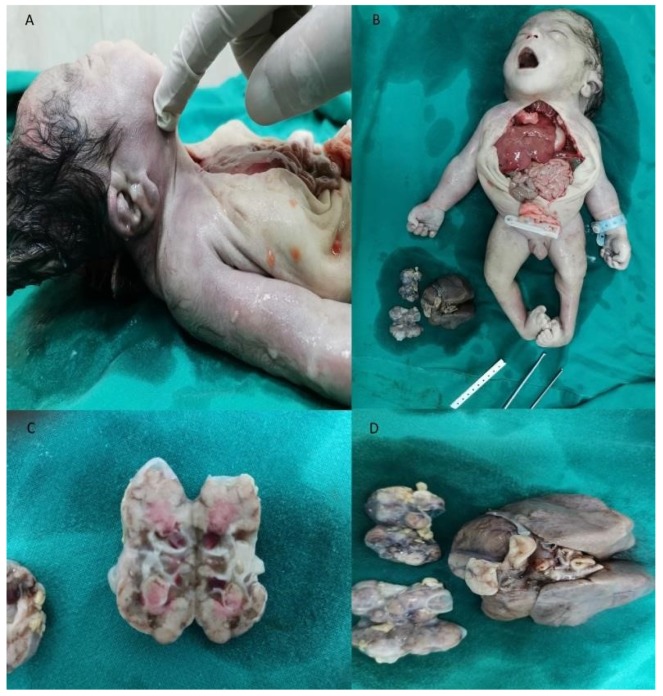
An autopsy of the affected twin. (**A**) Low set ears, preauricular sinus; (**B**) short neck, congenital bilateral talipes equinovarus, specific suborbital crease, kidneys and lungs (**C**) Gross appearance of polycystic kidneys; (**D**) Gross appearance of polycystic kidneys and hypoplastic lungs.

**Figure 3 medicina-56-00109-f003:**
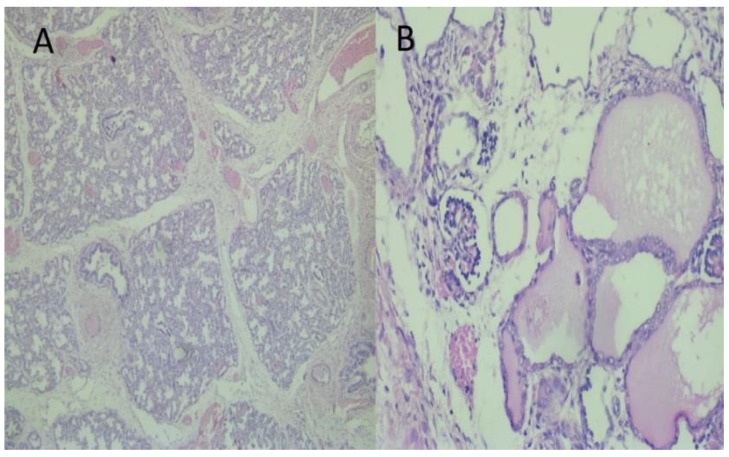
Histopathological examination of lungs and kidneys: (**A**) Immature terminal saccular phase of development of lung parenchyma. HE × 100; (**B**) Kidney with an area of primitive glomeruls and cysts as lined by cuboidal or flattened cells from dilated tubuls with eosinophilic content. HE × 100.

**Table 1 medicina-56-00109-t001:** Subtypes of Potter’s syndrome [4,5].

Classic Syndrome	Bilateral Renal Agenesis
Potter’s syndrome type 1	Autosomal recessive polycystic kidney
Potter’s syndrome type 2	Renal dysplasia
Potter’s syndrome type 3	Autosomal dominant polycystic kidney
Potter’s syndrome type 4	Obstruction either in kidney or ureter leading to kidney disease

**Table 2 medicina-56-00109-t002:** Potter’s sequence in monoamniotic pregnancy cases.

	McNamara et al. Case 1 [6]	McNamara et al. Case 2 [6]	Mauer et al. [7]	Cilento et al. [8]	Perez-Brayfield et al. [9]
**Prenatal complications**	Preterm labor due to premature rupture of membranes	Preeclampsia	Bleeding in the first trimester	Gestational diabetes, hypertension	Preterm labor
**Prenatal USG**	Twin B—bilateral multicystic kidneys	Twin B—ascites, dilated right renal pelvis	Not reported	Twin A—tetralogy of Fallot, bladder and right kidney absence; left kidney—dysplastic	No abnormalities found
**Type of delivery**	Cesarean delivery	Cesarean delivery	Vaginal delivery	Cesarean delivery	Vaginal delivery
**Gestational age of neonates**	33 weeks of gestation	29 weeks of gestation	38 weeks of gestation	36 weeks of gestation	35 weeks of gestation
**Sex of babies**	Males	Females	Males	Males	males
**Birth weight**	Twin A—1955 g Twin B—1575 g	Twin A—1320 g Twin B—1305 g	Twin A—2250 g	Twin A—1470g Twin B—1600g	Not reported
**Apgar score**	Not reported	Not reported	Not reported	Not reported	Both—10
**The affected twin extrauterine life**	Died 7 days after delivery	Discharged at 8 months of age	Died 12 days after delivery	Died 2 days after delivery	Died 2 months after delivery
**Physical/Autopsy findings of the affected twin**	VATER association	Cloacal dysgenesis	Twin A—low-set ears; absence of urethral meatus, kidneys, renal arteries, ureters	Bladder, right renal, gonadal agenesis, esophageal atresia	CT of twin A revealed—renal agenesis. Autopsy was not performed
**RDS**	No	No	No	No	No
**Genetic examination**	Not performed	Not performed	Not performed	Not performed	Not performed
**Family history**	Not applicable	Not applicable	Sibling of the twins had hypospadias with chordee	Not applicable	Not applicable

USG—ultrasonography; VATER—vertebral defects (V), anal atresia (A), tracheoesophageal fistula with esophageal atresia (TE), renal, and radial dysplasia (R); RDS—respiratory distress syndrome; PR—preterm labor.

**Table 3 medicina-56-00109-t003:** Potter’s sequence in diamniotic pregnancy cases.

	Sutherland [11]	Holden et al. Case 1 [12]	Holden et al. Case 2 [12]	Holden et al. Case 3 [12]	Fadel and Fulcher [10]	Presented Case
**Zygocity**	Not reported	Not reported	Not reported	Not reported	Not reported	Dizygotic
**Chorionicity**	Monochorionic	Monochorionic	Monochorionic	Dichorionic	Not reported	Dichorionic
**Prenatal complications**	No	Intrauterine growth retardation	No	Preterm labor due to SF	Preterm labor	PPROM
**Prenatal USG**	Twin B—oligohydramnios, bilateral hydroureteronephrosis, dilated bladder	Twin B—anhydramnios, bilateral hydronephrosis, and echogenic kidneys	Twin B—anhydramnios, bilateral hydronephrosis, megacystis	Twin B—anhydramnios, multicystic kidneys, urinary ascites	Twin B—oligohydramnios, bladder, and kidney absence	Not applicable
**Type of delivery**	Not reported	Vaginal delivery	Elective Cesarean section	Cesarean section	Vaginal delivery	Cesarean section
**Gestational age of neonates**	37 weeks	36 weeks	36 weeks	32 weeks	34 weeks	34 weeks
**Sex of babies**	Not reported	Females	Males	Females	Males	Males
**Birth weight**	Not reported	Twin A—2150 g Twin B—2140 g	Twin A—2510 g Twin B—1500 g	Twin A—1550 g Twin B—1700 g	Twin A—2325 g Twin B—984 g	Twin A—1690 g Twin B—1350 g
**Apgar score**	not reported	Both—normal	Not reported	Twin A—6, 9	Not reported	Twin A—2,3 Twin B—3,5
**The affected twin extrauterine life**	Discharged after 4 months of age	Died 2 days after birth	Died 3 h after birth	Twin B—selective feticide	Twin B—died shortly after birth	Died 3 hours after birth
**Physical or autopsy findings**	Mild nasal and ear compression	Bilateral ureteropelvic obstruction, dysplasia kidneys, severe PH	Bilateral renal dysplasia, Potter’s facies, severe PH	Urethral agenesis, severe PH	Autopsy refused	PH, PS facies, talipes equinovarus, polycystic kidneys
**Lung/body weight ratio**		Not reported	Not reported	Not reported	Not reported	0.008
**Microscopic findings**		Not reported	Not reported	Not reported	Not reported	Reported
**Genetic examination or family history**	Not performed	Normal karyotype	Normal karyotype	Not performed	No birth defects in family	Refused/no family birth defects

PPROM—Preterm prelabor rupture of amniotic membranes; PH—Pulmonary hypoplasia; PS—Potter’s syndrome; USG—ultrasonography; SF—selective feticide.

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
