# Peer review of "Discordance for Potter’s Syndrome in a Dichorionic Diamniotic Twin Pregnancy—An Unusual Case Report"

_medicina, 2020, doi:10.3390/medicina56030109_

Round 1
Reviewer 1 Report
Remove "rare" as it is a very generalized terminology.
"A 21-year-old female multigravida (gravida 2, parity 2, abortion-0) with 34 weeks", edit as "at 34 weeks..."
"Her first pregnancy through vaginal delivery was four years ago - she gave birth to a baby with Potter's syndrome" Reedit this sentence.
"Vaginal examination confirmed cephalic presentation in twin A...". Remove; it may be true the opposite that is an US confirmation of a clinical diagnosis!
"We performed a cesarean section due to twin A exhibiting signs of fetal distress..." Very apsecific definition. How did you diagnose "fetal distress"? Specify in detals. It was the anhidramnios, Doppler, CTG or both?
"The amniotic membrane of twin B was spontaneously open" Unusual terminology in Obstetrics practice. Specify and reedit terminology.
Authors refer to Apgar score and NOT to the only diagnostic test that is acid-base assessment on both umbilical artery and vein!!
"He had typical facial feautures for Potter's sequence – a flattened nose, low set ears, a specific suborbital crease and a short neck" Already states few sentences above. Duplicate.
Change "rare" and reedit as "uncommon", "unusual"...
When citing more than one author, "et. al" must be added.
"He requested for expectant management in twin pregnancies discordant for major (lethal) anomalies [20]" What he requested? Expectant management? If so, clearly specify.
If you would like to use acronymous, for example Potter's syndrome (PS) it should be used as the first time and then repeated throuout the Text.
Add a space following the Table and the Text.
"The manifestations of extrarenal features of PS depends on the amniocity. Case studies show that in monoamniotic twin pregnancy discordance for PS the affected twin has no extrarenal features. The rare cases where we have a discordance for Potter’s syndrome in diamniotic pregnancy are of particular interest as there is no consensus whether the unaffected twin protects the other by producing amniotic fluid and by decreasing the intrauterine pressure.". It is author's conclusion or opinio? If not, a Ref/s should be added.
Please add this Reference: Tonni G, Azzoni D, Ventura A, Ambrosetti F, De Felice C. Fetal Ped Pathol 2008;27(6):264-273.
Reviewer 2 Report
The authors reported the case of discordance for Potter’s syndrome in a dichorionic diamniotic twin pregnancy. They performed an autopsy and calculated lung weight/body weight ratio, as well as histological examination of lungs and kidneys.
The authors concluded that the appearance of external features in the affected twin depends on the amniocity. It is well known that the manifestations of external features of Potter’s syndrome depends on the amniocity. This paper is not informative or contributory.
Major
The first child also had Potter's syndrome, a genetic predisposition may have affected it. Genetic study is required. The much more detail information of the first child with Potter’s syndrome may be essential. In the discussion, previous case reports have been vaguely listed but not organaized. Previous reports of the information of Potter’s syndrome in twin pregnancy should be summarize in Table. Even in rare case reports, reports should be include new findings. How was the S/D value of umbilical artery blood flow by ultrasonography?
Minor
Introduction parts contained Case presentation. Introduction parts should be organized and simplified. A dichorionic diamniotic (DCDA) twin pregnancy account for all dizygotic pregnancies and 20% of monozygotic pregnancies. Which type was the case? The value of normal lung/body weight ratio at term and 34 weeks is required in the manuscript. Even after the abbreviation is once defined PS, the full names are repeated and mixed. (Potter’s sequence, Potter’s syndrome and PS) Page 2, Next to last line: 0,008 should be 0.008.
Round 2
Reviewer 2 Report
The revised version is improved compared with the original one.
Authors could add the limitation of the manuscript.
